# Overestimating women's representation in medicine: a survey of medical professionals' estimates and their (un)willingness to support gender equality initiatives

Christopher T Begeny ,[1] Rebecca C Grossman ,[2,3] Michelle K Ryan[1,4]

¹Psychology, University of Exeter, Exeter, UK
²NIHR Oxford Biomedical Research Centre, John Radcliffe Hospital, Oxford, UK
³Oxford Centre for Diabetes, Endocrinology and Metabolism, University of Oxford, Oxford, UK
⁴Global Institute for Women's Leadership, The Australian National University, Canberra, Australian Capital Territory, Australia

**Correspondence to**
Dr Christopher T Begeny;
C.Begeny@exeter.ac.uk

## ABSTRACT

**Objective** Amidst growing numbers of women in certain areas of medicine (eg, general practice/primary care), yet their continued under-representation in others (eg, surgical specialties), this study examines (1) whether medical professionals mistakenly infer that women are now broadly well represented, overestimating women's *true* representation in several different areas and roles; and (2) whether this overestimation of women's representation predicts decreased support for gender equality initiatives in the field, in conjunction with one's own gender.

**Design** Cross-sectional survey.

**Setting** UK-based medical field.

**Participants** 425 UK medical consultants/general practitioners and trainees (ST/CT1+/SHO/Registrar); 47% were female.

**Main outcome measures** Estimates of women's representation in different areas/roles within medicine, examined as a composite estimate and individually; and a multi-item measure of support for gender-based initiatives in medicine.

**Results** Medical professionals tended to overestimate women's true representation in several different areas of medicine (general practice, medical specialties, surgical specialties) and in various roles (consultants/general practitioners, trainees, medical school graduates). Moreover, these erroneous estimates predicted a decreased willingness to support gender-based initiatives, particularly among men in the field: composite overestimation*respondent gender interaction, $B=-0.04$, 95% CI −0.07 to −0.01, p=0.01. Specifically, while female respondents' (over)estimates were unrelated to their level of support ($B=0.00$, 95% CI −0.02 to 0.02, p=0.92), male respondents' tendency to overestimate the proportion of women in medicine predicted lower support for gender-based initiatives ($B=-0.04$, 95% CI −0.06 to −0.02, p<0.001).

**Conclusions** While some progress has been made in gender representation in the medical field, this research illustrates that there are still barriers to gender equality efforts and identifies who within the field is focally maintaining these barriers. It is those individuals (particularly men) who overestimate the *true* progress that has been made in women's representation who are at highest risk of undermining it.

### Strengths and limitations of this study

⇒ With women now well represented in some areas of medicine yet under-represented in others, there remains a dearth of evidence as to whether medical professionals are able to accurately gauge women's representation in different areas/roles; this study is designed to help fill this gap in knowledge.

⇒ There is also no known evidence as to whether the tendency to overestimate women's true representation can help explain why some medical professionals are reluctant to support gender equality initiatives in the field; this study is also designed to help fill this gap in knowledge.

⇒ The design of this research further enables us to help medical professionals and related organisations, as well as policymakers, identify barriers to gender equality efforts by identifying who within the field may be most likely to resist or withhold support for initiatives that aim to promote gender equality in the field.

⇒ More broadly, amidst ongoing efforts to promote gender equality in the medical field, the design of this research allows us to illustrate that it is important not only to consider the true representation of women in the field but also medical professionals' *perceptions* of women's representation.

⇒ This study was not designed to assess *why* some medical professionals' estimates of women's representation are linked to their level of support for gender equality initiatives.

## INTRODUCTION

Paralleling trends in other countries, in the UK women now make up over half of all medical school graduates.[1] [2] However, recruitment of female doctors to several specialty areas is not keeping pace with their recruitment to medicine in general.[3] [4] For instance, women are well represented in general practice/primary care, yet remain under-represented in medical and surgical

specialties (eg, in surgical specialties, less than 15% of consultants are women).[5]

Despite women's continuing under-representation in several areas of medicine (including some of the highest paying and most prestigious areas),[6–8] their more prominent representation in general practice and medical schools may be prompting some in the field to mistakenly infer that women are now well represented across the board or better represented than they actually are in several areas. This is important to consider, partly because if individuals *overestimate* women's representation they may be less willing to support policies and initiatives that aim to further promote gender equality in the profession. They may regard them as no longer necessary, for instance. Indeed, previous research on this topic, although limited in scope, demonstrates that when individuals overestimate women's representation in a field (eg, in STEMM (science, technology, engineering, mathematics and medicine) and in politics), they show less support for initiatives that aim to help women in those fields.[9–11] Thus, medical professionals who overestimate the true progress that has been made in women's representation in the field may be at highest risk of undermining it.

Medical professionals' tendency to support gender equality initiatives may hinge on more than their (over) estimates of women in the field, however. It may also depend on medical professionals' own gender. This is partly because gender-based initiatives and related groups (eg, the General Medical Council Gender Equality Scheme, Women in Surgery at the Royal College of Surgeons) aim to promote not just the representation of women but also the *equal treatment* of women—a recognition that true gender equality is achieved, and fundamentally defined, not just by numerical representation but the absence of gender bias in how women (and individuals of all genders) are perceived and treated. Thus, representation aside, individuals may continue supporting these gender-based initiatives if they are cognizant of ongoing issues with gender bias and discrimination in the field.[8 12–14] Indeed, recent evidence demonstrates that even when women become well represented in a field, gender biases and unequal treatment persist, and it is predominantly *women* in the field who remain cognizant of this fact (at significantly higher rates than men).[15] Ultimately, this suggests women in the medical profession may more reliably support gender-based initiatives, regardless of their estimations of women's numerical representation in the field, because they are more likely to see the ongoing value in these initiatives for combating gender bias. By comparison, because men are less likely to recognise issues of gender bias, their support for gender equality initiatives may more simply, and systematically, vary as a function of their tendency to overestimate women's representation.

## Current research

The current research examines whether medical professionals tend to overestimate women's representation in medicine and whether such erroneous estimates (along with their own gender) predict a decreased willingness to support gender-based initiatives. Using a sample of UK medical professionals, we first test whether individuals are generally accurate in estimating women's representation in different areas of medicine—general practice, medical and surgical specialties—and in different roles—consultants/general practitioners (GPs), trainees/junior doctors and medical school graduates. We then test whether, as hypothesised, overestimating women's representation predicts decreased support for gender-based initiatives and whether this is moderated by medical professionals' own gender.

## Gender stereotypical beliefs about women in medicine

As an exploratory step, we also examine individuals' endorsement of a gender stereotypical belief in men's superiority in the medical profession (eg, that men are simply better suited for the profession)—a belief that implies women should not be afforded equality in the profession and thus should predict lower willingness to support gender equality initiatives.[16 17] Assessing this belief therefore offers two potential insights. First, it allows us to test our core hypothesis—that overestimating women's representation predicts less support for gender-based initiatives, primarily among men—more conservatively by testing whether this effect (overestimation*respondent gender interaction) is robust even when accounting for the role of this belief in explaining individuals' (lacking) support for gender-based initiatives. Second, it allows us to assess whether there might be some men, like some women in medicine, who overestimate women's representation yet maintain a consistent level of support for these initiatives. This may be the case among men who more strongly reject this belief (tested via an overestimation*respondent gender*gender stereotypical belief interaction).

## METHODS

### Participants and procedure

Participants were 425 UK-based consultants/GPs and trainees/junior doctors (grades: ST/CT1+/SHO/Registrar [Specialty Trainee/Core Trainee/Senior House Officer/Registrar]) in the medical field (47% female; $M_{age}$=42.63, SD=11.82; role: 13.9%/4.5% consultants/trainees in general practice, 24.6%/12.0% consultants/trainees in medicine, 7.9%/6.7% consultants/trainees in surgery, 7.4% foundation year 1/2 doctors, 23.0% other (eg, doctors in industry positions, doctors in psychiatry); for more detailed descriptions of these areas and roles within medicine, see refs 18 19). Respondents completed a brief survey online described as aiming to 'better understand individuals' perceptions of doctors within the UK medical profession'. We recruited participants via email, disseminated through listservs maintained by the 24 medical royal colleges and faculties, 214 National Health Service Trusts, and 46 medical subspecialty and social societies. We also recruited respondents via social media and a doctors-only web forum. Participation was voluntary (no remuneration). We excluded four

respondents because they indicated that they did not work (nor had worked) in the UK and three for illogical responses (stating that they believed 98%–100% of all consultants and trainees, across all areas, were female; final sample size, n=418; n=377–418 for all primary analyses; missing data: 0–25 cases for area/role-specific estimates of women's representation, 41 cases for measure of support for gender-based initiatives). Sensitivity analyses indicated sample size was generally adequate (based on lowest n, $\alpha$=0.05, 1-$\beta$=0.80; for detecting d≥0.14 in one-sample t-test, for detecting $f^2$≥0.02 based on $\Delta R^2$ for the addition of the overestimation*respondent gender interaction term). All data underlying the findings described in this article are available at the Center for Open Science.[20]

### Patient and public involvement

No patients were involved; neither patients nor the public were directly involved in the design, conduct, reporting or dissemination plans of this research.

### Measures

Respondents answered questions measuring the following key constructs and provided demographic information (eg, gender, age, general area/role in medicine).

### Estimates of women by area/role

To assess respondents' estimates of the proportions of women in different areas/roles, we asked 'What percentage of ___ do you think are female?' with the following inserted: GP doctors, trainee GP doctors (ST/CT1+/SHO/Registrar), consultant doctors in medical specialties, trainee doctors in medical specialties (ST/CT1+/SHO/Registrar), consultant doctors in surgical specialties, trainee doctors in surgical specialties (ST/CT1+/SHO/Registrar), and medical school graduates. Respondents answered each of these seven questions on a sliding scale from 0% to 100%. To calculate the degree to which participants underestimated or overestimated true proportions, we subtracted the actual proportion of women within each area/role (obtained statistics aligned to the time of data collection in 2017[21 22]) from respondents' estimate. Thus, positive values reflected overestimation.

### Support for gender-based initiatives in the profession

To assess support for initiatives designed to support women in the UK medical profession, after explaining that such initiatives exist and providing examples (eg, the General Medical Council Gender Equality Scheme, Women in Surgery at the Royal College of Surgeons), we asked respondents to indicate how much they (dis)agree that these types of initiatives are necessary, fair, excessive/'over the top' (reverse-scored) or put men at a disadvantage (reverse-scored). These four items were rated 1–7 (*strongly disagree–strongly agree*), reliable ($\alpha$=0.85) and averaged to form a composite.

### Gender stereotypical beliefs about women in medicine

To assess endorsement of a gender stereotypical belief about men's superiority in the medical profession, we asked respondents how much they (dis)agree that, for example, there is something about being a man that makes one better suited for the medical profession (adapted from Danbold and Huo[16]). These six items were rated 1–7 (*strongly disagree–strongly agree*), reliable ($\alpha$=0.80) and averaged to form a composite.

### Overview of statistical methods

All statistical analyses were conducted in SPSS v28 (pairwise deletion used as necessary). This included bivariate (zero-order, Pearson) correlations (see table 1), one-sample t-tests (test value=0; see tables 2 and 3), independent samples t-tests (see table 3 superscripts) and tests of interactions using linear (ordinary least squares) regression via the PROCESS macro in SPSS, with 5000 resamples for generating percentile bootstrap CIs (for more details about PROCESS, see Hayes[23]). Primary regression analyses tested whether respondents' support for gender-based initiatives varied as a function of their tendency to overestimate the proportion of women in medicine and their own gender (overestimation*respondent gender interaction) using PROCESS model 1 (outcome: support for gender-based initiatives; predictor: overestimation of women's representation (mean-centred); moderator: gender (0 *female*, 1 *male*; mean-centred); covariate: age; analyses without covariate evinced the same statistically significant results). Follow-up regression analyses mirrored primary regression analyses while further testing whether the hypothesised overestimation*respondent gender effect was robust and/or qualified by respondents' endorsement of the gender stereotypical belief that men are superior in the medical profession (overestimation*respondent gender*gender stereotypical belief) using PROCESS model 3 (regression model identical to the primary regression model, but with the inclusion of a second moderator: endorsement of gender stereotypical belief, and its corresponding interaction terms).

### RESULTS

Table 1 provides bivariate correlations illustrating how female and male medical professionals' tendency to overestimate women's representation in a given area/role corresponds to their overestimations in other areas/roles, as well as their endorsement of gender stereotypical beliefs and support for gender-based initiatives.

### Respondents' estimates versus actual proportions of women by area/role

We first examined how respondents' estimated proportions of women in different areas/roles compared with actual proportions. Across areas, both male and female respondents tended to overestimate the proportion of female consultants and GPs. Estimated proportions of female trainees varied more by area. As noted in table 2,

**Table 1** Bivariate (zero-order) correlations by gender, with correlations among female and male respondents above and below the diagonal, respectively

| Variable | 1 | 2 | 3 | 4 | 5 | 6 | 7 | 8 | 9 |
|---|---|---|---|---|---|---|---|---|---|
| (Over) estimated % of female | | | | | | | | | |
| 1. TRs, general practice | – | 0.48*** | 0.39*** | 0.49*** | 0.30*** | 0.19** | 0.61*** | 0.13 | 0.11 |
| 2. TRs, medicine | 0.51*** | – | 0.39*** | 0.26*** | 0.42*** | 0.26*** | 0.43*** | 0.07 | 0.05 |
| 3. TRs, surgery | 0.20** | 0.27*** | – | 0.32*** | 0.41*** | 0.55*** | 0.45*** | 0.10 | −0.04 |
| 4. DRs, general practice | 0.64*** | 0.48*** | 0.11 | – | 0.40*** | 0.12 | 0.33*** | 0.14 | 0.05 |
| 5. DRs, medicine | 0.21** | 0.45*** | 0.30*** | 0.38*** | – | 0.53*** | 0.35*** | 0.04 | −0.05 |
| 6. DRs, surgery | 0.16* | 0.27*** | 0.46*** | 0.25*** | 0.52*** | – | 0.19** | 0.05 | −0.17* |
| 7. Medical school graduates | 0.61*** | 0.48*** | 0.09 | 0.53*** | 0.18** | 0.15* | – | 0.04 | 0.13 |
| 8. Gender stereotypical beliefs | 0.07 | 0.11 | 0.00 | 0.04 | 0.05 | 0.08 | 0.05 | – | −0.28*** |
| 9. Support for gender initiatives | −0.15* | −0.17* | −0.06 | −0.14 | −0.16* | −0.18** | −0.15* | −0.57*** | – |

The numbering across the top row of the table (1–9) corresponds to the variables, as numbered, in the left column.
*P<=0.05, **P<=0.01, ***P<=0.001
DRs, general practitioners/consultant doctors; TRs, trainee/junior doctors (ST/CT1+/SHO/Registrar [Specialty Trainee/Core Trainee/Senior House Officer/Registrar]).

these results were also largely evident (among both male and female respondents) when limiting analyses for a given area to the respondents who were themselves in that particular area of medicine. The results also showed that both male and female respondents overestimated the proportion of female medical school graduates (see table 3 for the results separated by respondent gender).

Tables 2 and 3 also show the SD for each mean estimated proportion. These highlight that, irrespective of the estimated proportion of women in an area/role *on average*, there was substantial variability in estimates within the sample of respondents. This variability is key to assessing whether these (over)estimations reliably predict individuals' (lower) levels of support for gender-based initiatives.

### Support for gender-based initiatives
To test whether respondents' support for gender-based initiatives varied by their tendency to overestimate the proportion of women in medicine and their own gender, we ran tests of interactions via PROCESS (model 1; see the Overview of statistical methods section for more details). Given that the measure of support for gender-based initiatives was not tied to one specific area or role within medicine, it is arguably most relevant to assess how respondents' levels of support varied as a function of their

**Table 2** Respondent estimates versus actual proportions of women by area/role

| Role | Area | Estimated % female (SD) | Actual % female | Difference (estimated–actual) | | | |
|---|---|---|---|---|---|---|---|
| Consultants/GPs | General practice | 58.25 (11.49) | 54 | 4.25 (3.15 to 5.36) | t=7.57 | P<0.001* | d=0.37 |
| | Medicine | 43.27 (11.15) | 37 | 6.27 (5.20 to 7.34) | t=11.50 | P<0.001* | d=0.56 |
| | Surgery | 24.99 (10.65) | 14 | 10.99 (9.97 to 12.02) | t=21.10 | P<0.001* | d=1.03 |
| Trainees | General practice | 63.55 (12.35) | 69 | −5.45 (−6.68 to −4.23) | t=−8.75 | P<0.001 | d=0.44 |
| | Medicine | 53.82 (10.15) | 53 | 0.82 (−0.19 to 1.83) | t=1.60 | P=0.11* | d=0.08 |
| | Surgery | 37.37 (11.91) | 33 | 4.37 (3.19 to 5.55) | t=7.27 | P<0.001* | d=0.37 |
| Medical school graduates | | 59.68 (9.83) | 55 | 4.68 (3.70 to 5.65) | t=9.44 | P<0.001* | d=0.48 |

Positive difference scores indicate overestimations of women's representation.
Values in brackets are 95% CIs around that difference score.
The t, p and d values indicate whether that difference score deviated significantly from 0 (one-sample t-test, effect size d; ie, whether estimations of women's representation significantly differed from their true representation).
*Virtually identical results evident (for both male and female respondents) when limiting analyses to respondents (trainees and consultants/GPs) who were themselves in this area of medicine (analyses not applicable regarding medical school graduates). Actual percentages reflect statistics aligned to the time of data collection (obtained from refs 21 22).
GPs, general practitioners.

**Table 3** Respondents' estimates versus actual proportions of women by area/role, examined separately for male and female respondents

| Role | Area | Estimated % female (SD) | Estimated by | Actual % female | Difference (estimated−actual) | | | |
|---|---|---|---|---|---|---|---|---|
| Consultants/GPs | General practice | 56.83 (11.14) | Estimated by Male respondents | 54 | 2.83ᵃ (1.35 to 4.31) | t=3.77 | P<0.001 | d=0.25 |
| | | 59.83 (11.69) | Female respondents | | 5.83ᵃ (4.19 to 7.47) | t=7.01 | P<0.001 | d=0.50 |
| | Medicine | 42.76 (10.61) | Estimated by Male respondents | 37 | 5.76 (4.35 to 7.17) | t=8.06 | P<0.001 | d=0.54 |
| | | 43.83 (11.72) | Female respondents | | 6.83 (5.19 to 8.48) | t=8.21 | P<0.001 | d=0.58 |
| | Surgery | 24.75 (10.62) | Estimated by Male respondents | 14 | 10.75 (9.34 to 12.17) | t=15.02 | P<0.001 | d=1.01 |
| | | 25.26 (10.71) | Female respondents | | 11.26 (9.76 to 12.76) | t=14.79 | P<0.001 | d=1.05 |
| Trainees | General practice | 62.28 (11.91) | Estimated by Male respondents | 69 | −6.72ᵇ (−8.36 to −5.08) | t=−8.08 | P<0.001 | d=0.57 |
| | | 64.93 (12.70) | Female respondents | | −4.07ᵇ (−5.90 to −2.24) | t=−4.39 | P<0.001 | d=0.32 |
| | Medicine | 53.15 (10.28) | Estimated by Male respondents | 53 | 0.15 (−1.27 to 1.56) | t=0.20 | P=0.84 | d=0.01 |
| | | 54.55 (9.99) | Female respondents | | 1.55 (0.12 to 2.99) | t=2.13 | P=0.03 | d=0.16 |
| | Surgery | 37.36 (11.48) | Estimated by Male respondents | 33 | 4.36 (2.78 to 5.94) | t=5.43 | P<0.001 | d=0.38 |
| | | 37.38 (12.40) | Female respondents | | 4.38 (2.59 to 6.16) | t=4.84 | P<0.001 | d=0.35 |
| Medical school graduates | | 59.75 (8.48) | Estimated by Male respondents | 55 | 4.75 (3.58 to 5.92) | t=8.02 | P<0.001 | d=0.56 |
| | | 59.60 (11.13) | Female respondents | | 4.60 (2.99 to 6.20) | t=5.66 | P<0.001 | d=0.41 |

Positive difference scores indicate overestimations of women's representation.

Values in brackets are 95% CI around that difference score.

The t, p and d values indicate whether that difference score deviated significantly from 0 (one-sample t-test, effect size d; ie, whether estimations of women's representation significantly differed from their true representation).

The superscripts 'a' and 'b' indicate the magnitude of male and female respondents' overestimation/underestimations (ie, their mean deviations from the actual % female) for this area/role significantly differing from one another (t=2.68/2.14, p=0.01/.03, d=0.26/.22). For all other areas/roles (without a superscript), male and female respondents' overestimations did not significantly differ from one another (all t≤1.37, p≥0.17). Actual percentages reflect statistics aligned to the time of data collection (obtained from refs 21 22).

GPs, general practitioners.

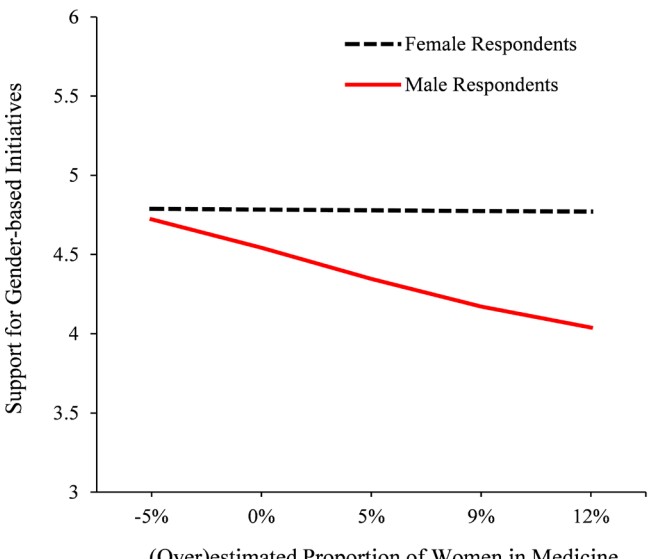

**Figure 1** Male and female respondents' (ie, medical professionals) support for gender-based initiatives in the UK medical profession (1–7 scale) as a function of their estimates of the proportion of women in medicine. Positive values on the x-axis reflect an overestimation of women's representation. Female respondents' estimates were unrelated to their level of support ($B$=0.00, 95% CI −0.02 to 0.02, p=0.92). By comparison, male respondents' tendency to overestimate the proportion of women in medicine predicted significantly less support for gender-based initiatives ($B$=−0.04, 95% CI −0.06 to −0.02, p<0.001; overestimation*respondent gender interaction, $B$=−0.04, 95% CI −0.07 to −0.01, p=0.01, $\Delta R^2$=0.02 for the addition of interaction term, $F(1,372)$=6.48, p=0.01, $f^2$=0.02).

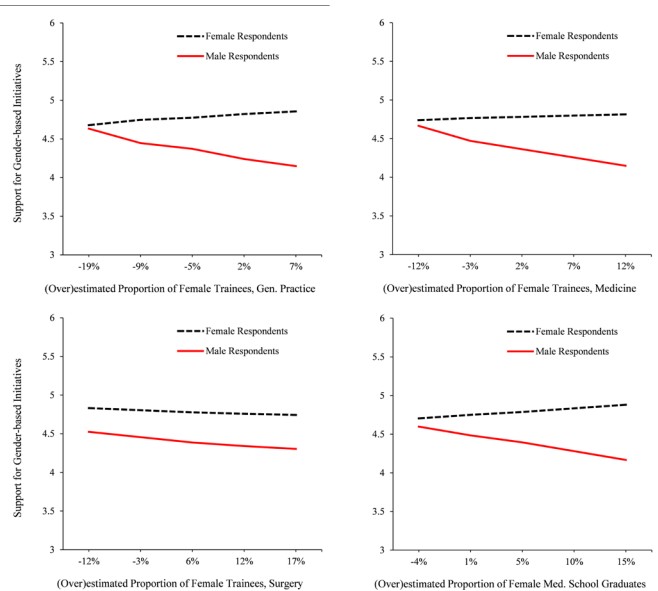

**Figure 2** Male and female respondents' (ie, medical professionals) support for gender-based initiatives in the UK medical profession (1–7 scale) as a function of their estimates of the proportion of (1) female trainees in general practice, (2) medicine and (3) surgery and (4) female medical school graduates. Positive values on the x-axis reflect an overestimation of women's representation in that area/role. In the areas of general practice and medicine, and regarding medical school graduates, female respondents' estimates were unrelated to their level of support, yet male respondents' tendency to overestimate the representation of women in these areas/roles predicted significantly less support for gender-based initiatives. In surgery, neither women's nor men's estimates of female trainees predicted level of support.

*overall* tendency to overestimate women's representation (aggregated across areas/roles). We therefore computed a composite score (M=3.84, SD=7.47) reflecting respondents' average tendency to overestimate women's representation across the seven aforementioned areas/roles (α=0.80 for the seven estimated areas/roles).

As figure 1 shows, the results evinced differences in support for gender-based initiatives as a function of respondents' tendency to overestimate the proportion of women in medicine and their own gender (overestimation*respondent gender interaction, $B$=−0.04, 95% CI −0.07 to −0.01, p=0.01, $\Delta R^2$=0.02 for the addition of the interaction term, $F(1,372)$=6.48, p=0.01, $f^2$=0.02; overall $F(4,372)$=8.53, p<0.001; overestimation, $B$=−0.02, 95% CI −0.04 to −0.01, p=0.01; respondent gender, $B$=−0.40, 95% CI −0.65 to −0.16, p=0.001). Generally speaking, this means that as medical professionals got more severe in their overestimations of women's true representation, the disparity between female and male medical professionals' support for gender-based initiatives grew larger, as illustrated in figure 1.

Tests of simple slopes further showed that female respondents' (over)estimates were unrelated to their level of support ($B$=0.00, 95% CI −0.02 to 0.02, p=0.92),

yet male respondents' tendency to overestimate the proportion of women in medicine predicted lower support for gender-based initiatives ($B$=−0.04, 95% CI −0.06 to −0.02, p<0.001). In other words, among female respondents, regardless of their estimations of women in medicine, there was no systematic difference in their level of support for gender-based initiatives. Yet, among male respondents, there were systematic differences; in essence, for every 1% increase in their (over)estimations of the proportion of women in medicine, men's support for gender-based initiatives dropped by 0.04 points on average (thus, being 12% higher in one's overestimations equated to approximately a half-point decrease in level of support; see figure 1 for a visual illustration).

We also tested these interaction effects by area/role. As figure 2 shows, regarding estimates of female trainees in general practice, the results showed the same pattern of results (overestimation*respondent gender interaction, $B$=−0.03, 95% CI −0.05 to −0.01, p=0.01, $\Delta R^2$=0.02 for addition of interaction term, $F(1,372)$=7.13, p=0.01; overall $F(4,372)$=7.37, p<0.001). Simple slopes showed that female respondents' estimates of female trainees in this area were unrelated to their level of support ($B$=0.01, 95% CI −0.01 to 0.02, p=0.30), yet male respondents'

tendency to overestimate the proportion of women in this area predicted less support for gender-based initiatives (*B*=−0.02, 95% CI −0.03 to −0.01, p=0.01). This same pattern was also found regarding estimates of female trainees in medicine (overestimation*respondent gender interaction, *B*=−0.03, 95% CI −0.05 to −0.003, p=0.03; simple slopes: female respondents, *B*=0.00, 95% CI −0.01 to 0.02, p=0.71; male respondents *B*=−0.02, 95% CI −0.04 to −0.01, p=0.01), although not for surgery where, notably, women's representation is still quite low (overestimation*respondent gender interaction, *B*=0.00, 95% CI −0.02 to 0.02, p=0.65). Regarding estimates of female medical school graduates, the results again evinced a significant interaction (overestimation*respondent gender interaction, *B*=−0.03, 95% CI −0.07 to −0.01, p=0.01; simple slopes: female respondents, *B*=0.01, 95% CI −0.01 to 0.02, p=0.22; male respondents *B*=−0.02, 95% CI −0.04 to −0.003, p=0.02).

This same pattern of results was also evident when examining respondents' estimates of female GPs/consultants by area, although the effects were more modest (overestimation*respondent gender interactions: general practice, *B*=−0.02, 95% CI −0.04 to 0.00, p=0.06; medicine, *B*=−0.01, 95% CI −0.03 to 0.01, p=0.17; surgery, *B*−0.01, 95% CI −0.03 to 0.02, p=0.61). Again, in areas of general practice and medicine (not surgery), female respondents' estimates of female doctors in these areas were unrelated to their level of support (simple slopes for female respondents: general practice, *B*=0.00, 95% CI −0.01 to 0.02, p=0.81; medicine, *B* −0.01, 95% CI −0.02 to 0.01, p=0.35; surgery, *B*=−0.02, 95% CI −0.03 to 0.00, p=0.05). Yet male respondents' tendency to overestimate the proportion of female doctors in these areas predicted less support for gender-based initiatives (simple slopes for male respondents: general practice, *B*=−0.02, 95% CI −0.03 to −0.004, p=0.01; medicine, *B*=−0.02, 95% CI −0.04 to −0.01, p=0.01; surgery, *B*=−0.02, 95% CI −0.04 to −0.01, p=0.004).

## Follow-up analysis

In a follow-up analysis (PROCESS model 3; paralleling primary analysis using overestimation composite), we tested whether the hypothesised overestimation*respondent gender effect was robust and/or qualified by respondents' endorsement of the gender stereotypical belief that men are superior in the medical profession.

The results showed that those who more strongly endorsed this belief had less support for gender-based initiatives (gender stereotypical belief: *B*=−0.44, 95% CI −0.53 to −0.34, p<0.001; overestimation, *B*=−0.01, 95% CI −0.03 to 0.00, p=0.06; respondent gender, *B*=−0.34, 95% CI −0.55 to −0.13, p=0.001; overall *F*(8,362)=18.90, p<0.001). Yet, at the same time, the hypothesised overestimation*respondent gender interaction remained significant (*B*=−0.04, 95% CI −0.07 to −0.01, p=0.01). Thus, even when accounting for the role of individuals' endorsement of this belief, their level of support for gender-based initiatives still systematically varied by the tendency to overestimate the proportion of women in medicine and their

own gender. The results also showed that this interaction was not qualified by a three-way interaction (overestimation*respondent gender*gender stereotypical belief; *B*=−0.01, 95% CI −0.03 to 0.02, p=0.70), further illustrating its robustness in explaining individuals' support for gender-based initiatives.

While the three-way interaction was non-significant, the hypothesised effect at different levels of endorsement of this gender stereotypical belief did illustrate a potentially informative pattern of results. Specifically, male and female respondents who overestimated the proportion of women in medicine but also strongly *rejected* this belief (at the 25th percentile in the belief endorsement range) did not differ in their level of support for gender-based initiatives (*B*=−0.03, 95% CI −0.07 to 0.01, p=0.14): neither female (*B*=0.01, 95% CI −0.02 to 0.04, p=0.62) nor male (*B*=−0.02, 95% CI −0.05 to 0.01, p=0.11) respondents' tendency to overestimate the proportion of women in medicine predicted less support for initiatives. Yet, among those who more strongly *endorsed* this belief (at the 75th percentile), male and female respondents did differ in their support (*B*=−0.04, 95% CI −0.07 to −0.01, p=0.01): female respondents' overestimates were unrelated to support (*B*=0.00, 95% CI −0.02 to 0.03, p=0.78), while male respondents' overestimates predicted less support for gender-based initiatives (*B*=−0.04, 95% CI −0.06 to −0.02, p=0.001), such that among male respondents who more strongly endorsed this belief every 1% increase in their (over)estimations of women in medicine equated to an average 0.04 point drop in support for gender-based initiatives. Thus, while these analyses were exploratory, they suggest that men who overestimate women's representation may not be invariably more reluctant to support gender-based initiatives. There may be a subset of men who, despite overestimating women's representation, maintain a level of support for gender-based initiatives on par with that of their female counterparts, specifically those men who more strongly reject the gender stereotypical belief that men are more suitable for the profession.

## DISCUSSION

The strength and quality of the medical profession, including its ability to address an array of public health issues and ensure patient satisfaction, hinge on recruiting, retaining and supporting the full range of diverse talent that exists in the population, including among women.[14 24] In this vein, various initiatives are underway to increase women's representation in medicine, with some signs of progress.

Yet, amidst this growing gender diversity in medicine, with women now well represented in some areas yet under-represented in others, it is important to understand how medical professionals are perceiving this changing demographic landscape. The current research shows that amidst growing numbers of women, medical professionals are tending to overestimate women's true representation, with adverse implications. This research

shows that when individuals, particularly men, overestimate the proportion of women in medicine they express less support for gender-based initiatives that are striving to promote greater equality. Thus, men who overestimate the *true* progress that has been made in women's representation are at highest risk of undermining it.

This points to an insidious consequence that can arise when women's representation grows within a given field. It seems to prompt some to misperceive and overstate the actual degree of change, and following from this, particularly for men, mistakenly infer that gender equality initiatives in the field are no longer worth supporting. This ultimately hinders efforts to promote true equality, whether it be promoting women's representation in areas of the field where they are still under-represented or combating issues of gender bias that exist independent of women's numerical representation.[15]

In practical terms, this research illustrates the very real nature of the issue—that medical professionals are indeed overestimating women's representation in several areas and roles in the field. Simultaneously, it helps identify *who* within the field is at the highest risk of resisting efforts to promote gender equality.

This study does have its limitations. These include uncertainty around the total number of medical professionals who saw the study invitation (given methods for dissemination) and thus the response rate. Additionally, while this study examined estimates of women's representation across seven different key areas and roles, including GPs/consultants and trainees, future research might examine additional roles (eg, specialty and associate specialist doctors) or specialty areas.

The cross-sectional nature of these data precludes tests of causality. However, previous experimental work supports our hypothesised directionality of effect,[16] suggesting that when (male) medical professionals overestimate growth in the number of women in their field it results in less support for gender-based initiatives.

In future research, it will also be important to probe the mechanisms underpinning this overestimation effect. One possibility is that overestimating women's representation prompts individuals, particularly men, to genuinely although naïvely infer that gender bias is no longer an issue in their profession—believing that the biases and discrimination that once prevented women from entering the field are no longer occurring (see also refs 9 15). As a result, they may regard ongoing gender-based initiatives as unnecessary.

Another possibility is that overestimating women's representation predicts lower support for gender-based initiatives because that overestimation reflects a heightened sense of threat that some men feel, prompting them to exert more resistance to that changing demographic landscape (eg, expressing less support for gender-based initiatives).[16] Notably though, our overestimation*respondent gender effect held true when accounting for individuals' endorsement of the gender stereotypical belief that men are better suited for the medical profession. This

is important because research suggests endorsement of such a belief *reflects* men's sense of threat (ie, they endorse this type of belief when they feel their high status position in a profession is threatened).[17] In this way, it seems that an overestimation effect may stand independent of, or is at least not fully explained by, a sense of threat induced by a perceptible growth in women in the field.

Overall, this suggests multiple strategies may be required to address the consequences of this overestimation effect, depending on whether or for whom it is underpinned by a sense of threat versus naïveté about ongoing issues of under-representation (if not also ongoing issues of gender bias). For instance, targeted information campaigns that increase knowledge and awareness about women's true representation in different areas of medicine, along with information about persisting forms of gender bias (separate from matters of representation), may be useful in fostering greater support for gender-based initiatives among medical professionals whose reservations about these initiatives are rooted in genuine naïveté about persisting issues with under-representation and bias. Yet among those whose resistance is rooted in a sense of threat by growing proportions of women in the profession, other strategies may be necessary (eg, work-related self-affirmation techniques that alleviate this sense of threat).[25 26] There are a number of other potential strategies to consider as well, including those that aim to directly promote greater gender equality (for reviews, see refs 14 27).

It will also be important to consider whether there are thresholds for spurring this effect. In the current research, we found that while overestimations of women across most areas/roles predicted lower support for gender equality initiatives, this was not so for surgical specialties (both regarding estimates of GPs/consultants and trainees). This may be because both the actual representation and individuals' overestimations of women in this area are still relatively low (eg, actual and estimated proportions of female consultants in surgery: 14% and 25%; see table 2). This suggests that when it is still quite clear that women are vastly under-represented, aversion to gender equality initiatives is not piqued, perhaps either because it remains clear that those initiatives are still necessary (from the perspective of a 'naïve' overestimator) or because the still-low representation of women does not yet elicit threat (from the perspective of a 'threatened' overestimator).

It is also notable that medical professionals' endorsement of the gender stereotypical belief that men are better suited for the profession was unrelated to their tendency to overestimate the proportion of women in the field (see table 1). This held true for both male and female respondents. It suggests that overestimations of women's representation do not simply reflect a negative, pre-existing attitude (about women's suitability for the profession). Thus, while future research should further probe this relationship, their independence here indicates that medical professionals' estimates of women's

representation are, in their own right, an important basis for understanding who is likely to support gender equality initiatives or resist them, particularly among men in the profession. While endorsement of this gender stereotypical belief is important to consider, medical professionals' (over)estimations of women are key too.

Going forward, it will also be important to probe the role of gender in moderating the evinced overestimation effect. One possibility is that this gender-moderated effect reflects the fact that men are more likely than women to be unaware—or simply deny—that gender bias is still an issue in their profession (ie, in the most precise theoretical terms, it is one's belief that gender bias is no longer an issue, more than gender, that moderates the effect[15 28]). Another possibility is that this gender-moderated effect reflects an expression of ingroup favouritism[29 30]; if individuals perceive gender-based initiatives as generally beneficial to women (as a group) but not men, and they are motivated to act in ways that support their own gender-based ingroup (eg, because they highly identify with their gender), women may be generally supportive of these initiatives while men may not be, especially if men's overestimation of women in the field helps justify a belief that making deliberate efforts to support members of an outgroup is no longer necessary (ie, supporting initiatives that perceptibly benefit women).

Future research might also examine whether the general public similarly tends to overestimate women's representation in the medical profession. Individuals outside the profession would presumably be just as prone, if not more so, to these erroneous estimates. If so, given the current evidence that this has adverse implications for one's willingness to support gender equality initiatives, this would underscore the gravity of the issue, highlighting that resistance to establishing gender equality in the medical field may be coming from both those within and outside of the profession. In a similar vein, it will be valuable to examine whether these processes are evident specifically among leaders within the medical profession.

## CONCLUSION

Amidst ongoing efforts to promote greater gender equality in medicine, the current research illustrates that it is important not only to consider the true representation of women in the field, but also medical professionals' *perceptions* of women's representation. As shown, individuals' (mis)perceptions are accompanied by growing reservations, or less support for, gender equality initiatives. In this way, individuals' erroneous estimates mean less support for initiatives that are ultimately working to make the profession *truly* equitable for women.

**Correction notice** This article has been corrected since it was published Online First. The liecense has been updated to CCBY.

**Contributors** Conceptualisation: CTB, RCG, MKR. Data curation: CTB, RCG. Formal analysis: CTB. Funding acquisition: MKR. Methodology and design: CTB, RCG, MKR. Project management and administration: CTB. Visualisation: CTB. Writing, original draft: CTB, RCG. Writing, review and editing: CTB, RCG, MKR. Guarantor: CTB.

**Funding** This work was supported by a European Research Council Consolidator Grant (ERC-CoG 725128) awarded to MKR, and an MRC Fellowship grant (MR/T007974/1) awarded to RCG.

**Competing interests** None declared.

**Patient and public involvement** Patients and/or the public were not involved in the design, or conduct, or reporting, or dissemination plans of this research.

**Patient consent for publication** Not required.

**Ethics approval** This study involves human participants and was approved by and carried out in compliance with the standards for human research set forth by the University of Exeter Ethics Committee (approval for eCLESPsy000134). Informed consent was obtained from participants. Participants gave informed consent to participate in the study before taking part.

**Provenance and peer review** Not commissioned; externally peer reviewed.

**Data availability statement** Data are available in a public, open access repository. All data underlying the findings described in this article are available at the Center for Open Science (https://osf.io/hrm63/).

**ORCID iDs**
Christopher T Begeny http://orcid.org/0000-0003-4734-8840
Rebecca C Grossman http://orcid.org/0000-0003-1091-9275

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
