## [Reviewer comments · BMJ Open]

ARTICLE DETAILS

TITLE (PROVISIONAL)	Overestimating women's representation in medicine: A survey of medical professionals' estimates, and their (un)willingness to support gender-equality initiatives
AUTHORS	Begeny, Christopher; Grossman, Rebecca; Ryan, Michelle

VERSION 1 – REVIEW

REVIEWER	Fleming, Simon Queen Mary University of London, Institute of Health Sciences Education
REVIEW RETURNED	23-Jul-2021

GENERAL COMMENTS	It is one of life's pleasures to be able to accept a manuscript outright. When the paper in question addresses such an important and potentially nuanced topic with both qualitatively robust and culturally sensitive perspectives, it is even more so.
--

REVIEWER	Ni, H. Wenwen Sonoma State University, Psychology
REVIEW RETURNED	10-Aug-2021

GENERAL COMMENTS	I'd like to start by congratulating the authors on an interesting and clearly written paper. I hope that the feedback I provide below will prove useful as they finalize their publication. There are several points in the introduction that would benefit from greater precision and extrapolation. For example, the first paragraph mentions that "women are well represented in general practice / primary care, yet remain underrepresented in medical and surgical specialties" – it is important to mention that surgery and other medical specialties in which women are underrepresented tend to be seen as more prestigious. (See Crawley, 2014 for an example paper on the relationship between gender balance and occupational prestige). I also suggest that gender equality in the profession should be defined in the Introduction – i.e., what would be considered gender equality? Would it be 50% representation in all areas of medicine? Last of all, with regards to potential gender differences in supporting gender equality initiatives, there are likely a multitude of reasons in addition to gender bias. For example, gender-based initiatives usually benefit women but not men. That suggests that women will support them regardless of representation, while men may be looking for a reason not to support them. Overestimating women's representation may provide men with such a reason. In terms of the literature review, I was unclear about how this study fits into the existing literature regarding estimates of women's representation in medicine. Has there been past research on the overestimation of women in medicine, or is this study the first? What about past research on estimations of women in other professions? I
--

	would recommend adding a literature review on the effects of biased estimates of group representation. Overall, I found the Methods section very clear. I do feel the paper would benefit from definitions of each type of doctor in the study – for example, I am unclear about the distinction between general practice and medicine, or junior doctors and consultant doctors. In terms of Results, I think that Table 2 should be broken out by the gender of respondent. I was very curious about whether men or women were more likely to overestimate women’s representation in each area. I would also recommend breaking out the data by age of respondent and by respondent’s role – in particular whether they are a manager or a supervisor. There are potentially greater implications for the biased estimations of people in positions of power (who make hiring decisions, etc.). I am also interested in why the researchers think gender stereotypical beliefs did not correlate with overestimations – that seems counterintuitive so it would be good to have more discussion of this finding. In terms of future directions, I would be interested in the researchers’ views on what the data would look like if the respondents were from the general public, instead of medical professionals. What are the implications of the public overestimating the percentage of women in medicine vs. medical professionals’ overestimation? Last of all, some minor points:  • STEM should be written out the first time it is mentioned • In the first line of the current research section – “The current research examines medical professionals’ tendency to overestimate . . .” – I would add the word “whether” – the introduction shouldn’t be written as if overestimation is a given References: Crawley, D. (2014). Gender and perceptions of occupational prestige: Changes over 20 years. Sage Open, 4(1), 2158244013518923.
--	---

REVIEWER	Farkas, Amy H
REVIEW RETURNED	Medical College of Wisconsin, Division of General Internal Medicine 14-Oct-2021

GENERAL COMMENTS	This is a very interesting paper that highlights the role that perception versus reality plays in gender equity within medicine. This is a very well done study. The paper is well written with a clear description of their methodology and results. The discussion highlights the importance of this work. My only suggestion to the authors would be in their discussion they state that multiple strategies might be needed in order to overcome the issue of overestimation and I wonder if they could suggest some possible strategies to foster support for gender equity programs. Beyond that I do not have any additional suggestions.
---

REVIEWER	Holliday, Elizabeth
REVIEW RETURNED	The University of Newcastle, Australia, Rm4107, Lvl4W 02-Dec-2021

GENERAL COMMENTS	This is an interested and well written paper with fascinating conclusions. If these conclusions are valid, this paper would represent an important addition to the literature. While I greatly enjoyed reading the introduction, my review of the methods and results was hampered by the very limited description of statistical methods used. This made it difficult to understand how the analysis was performed, and how the results should be interpreted.
---

However, I believe these shortcomings can be readily rectified and have outlined suggestions for doing so below. I am willing to review a revised version of this manuscript that addresses my suggestions.

Methods

I found the reporting of statistical methods to be incomplete; I could find no “statistical methods” section in the methods. Please include such a section that describes all statistical tests used, and the software package. Then update the STROBE checklist to show where this content is provided.

For example, please the type of correlation statistic and how the correlation was estimated to assist with interpretation of results in Table 1 (or Table 2, as per my suggestion for a new Table 1 below). The meaning of the correlation values in Table 1 was unclear to me – which variables do these represent the correlation between?

On this same point, later in the results section there are interaction estimates presented, with confidence intervals and model fit statistics. Please describe the regression model used to estimate these terms, specifying the response variable and all explanatory variables. Also describe how any assumptions of the model were assessed. It is difficult to properly assess the results without this information.

I later noticed a brief description of a moderated regression analysis mentioned partway through the results section, but this content should be described in the methods, after first specifying the type of regression model used. Please describe the type of regression model using conventional terminology, that makes the assumed distribution for the response variable or errors clear, e.g., linear regression or a generalised linear model with the assumed response distribution and link function used.

For the interaction terms, I suggest you describe assessing “interaction” or “effect modification” rather than “moderator analysis” for consistency with the rest of the paper and with terminology widely used in epidemiology.

Results

Tables

It is standard to include a table showing participant characteristics for key variables that may relate to the exposure or response, e.g., as frequency with percent for categorical variables, and mean with SD or median with IQR for continuous variables. I suggest including this as a new Table 1 and renumbering the existing tables to follow. When describing results from the regression models, it would be helpful to offer the reader explicit interpretations of the beta coefficients and particularly the interaction terms at key instances (e.g., one explanation for each model). In addition to clearly describing the models (including outcomes and explanatory variables) in the methods, such interpretations will help the reader to understand the magnitude and relevance of the estimated effects. This is particularly important for interaction terms, which most readers do not intuitively understand. Spell it out – what does a positive or negative, significant interaction term mean in the context of each model? What does it imply about the modification of a particular effect by gender, for example?

VERSION 1 – AUTHOR RESPONSE

Reviewer: 1

Dr. Simon Fleming, Queen Mary University of London

Comments to the Author:

It is one of life's pleasures to be able to accept a manuscript outright. When the paper in question addresses such an important and potentially nuanced topic with both qualitatively robust and culturally sensitive perspectives, it is even more so.

Response #5

We really appreciate your supportive comments!

Reviewer: 2

H. Wenwen Ni

Comments to the Author:

I'd like to start by congratulating the authors on an interesting and clearly written paper. I hope that the feedback I provide below will prove useful as they finalize their publication.

Response #6

Thanks so much for all of your helpful comments. We have worked to address each of them, and the quality of the paper has surely grown as a result.

There are several points in the introduction that would benefit from greater precision and extrapolation. For example, the first paragraph mentions that "women are well represented in general practice / primary care, yet remain underrepresented in medical and surgical specialties" – it is important to mention that surgery and other medical specialties in which women are underrepresented tend to be seen as more prestigious. (See Crawley, 2014 for an example paper on the relationship between gender balance and occupational prestige). I also suggest that gender equality in the profession should be defined in the Introduction – i.e., what would be considered gender equality? Would it be 50% representation in all areas of medicine? Last of all, with regards to potential gender differences in supporting gender equality initiatives, there are likely a multitude of reasons in addition to gender bias. For example, gender-based initiatives usually benefit women but not men. That suggests that women will support them regardless of representation, while men may be looking for a reason not to support them. Overestimating women's representation may provide men with such a reason.

Response #7

We have made revisions in response to each of your aforementioned points, while also keeping advised word limits in mind. Specifically:

(1) In the Introduction, we now state that differences in women's representation across areas of medicine have some parallels to the differing levels of pay and prestige around those areas. Specifically, we now state (new text in bold):

"Despite women's continuing underrepresentation in several areas of medicine (**including some of the highest paying and most prestigious areas**) [6-8], their more prominent representation in general practice..."

(2) In the Introduction, we highlight for readers that gender equality is defined not just by numerical representation but the equal treatment of women (and individuals of other genders). Specifically, we now state (new text in bold):

"...gender-based initiatives and related groups (e.g., the General Medical Council Gender Equality Scheme, Women in Surgery at the Royal College of Surgeons) aim to promote not just the representation of women but also the *equal treatment* of women – **a recognition that true gender equality is achieved, and fundamentally defined, not just by numerical representation but the absence of gender bias in how women (and individuals of all genders) are perceived and treated.** Thus, representation aside, individuals may continue supporting..."

It is worth noting that with this revision we do not aim to offer a strict definition of gender equality that hinges on, for example, some fixed numerical threshold related to representation (e.g., 50%). This is in part because evidence suggests there can be varying conceptions of what constitutes 'equal' when it comes to numerical representation (e.g., Danbold & Unzueta, 2020), but also – perhaps more importantly – because what is most pertinent to the

current study is not how individuals (nor we as authors) integrate particular numerical thresholds into a definition of equality, but that numerical representation is not itself true gender equality (at any threshold). In other words, what is most important here is the idea that gender equality is about more than 'the numbers.' It is also about unbiased perceptions and treatment of women (and individuals of all genders), which is what we have aimed to highlight with this revision.

(3) In the Discussion section, we integrate your point that there could be other explanations for why men (who overestimate women's representation) may be less likely to support gender-equality initiatives. In addition to the possibility that men are less supportive of these initiatives because they are generally less likely to recognize that gender biases can still be an issue even after women's numerical representation has substantially grown (discussed in the Intro), and in addition to the possibility that men are less supportive of these initiatives because the recognition of women's growing numerical representation elicits a sense of threat (covered in the Discussion), as you say,

"...gender-based initiatives usually benefit women but not men. That suggests that women will support them regardless of representation, while men may be looking for a reason not to support them. Overestimating women's representation may provide men with such a reason."

If we follow your comment correctly, it suggests that social group/identity-based motives could also potentially be at play. For example, from the perspective of social identity theory and in line with notions of ingroup favoritism, individuals can be motivated to act in ways (or support actions/initiatives) that benefit their own ingroup. In this context, it could mean that if individuals perceive these gender-based initiatives as generally beneficial to women (as a group) but not men (as a group), women may generally be supportive of them while men will not be.

Overall, we think this fits nicely with some other ideas covered in the Discussion about the role of gender in moderating this overestimation effect. Therefore, in the Discussion we now state (all text is new):

"Another possibility is that this gender-moderated effect reflects an expression of ingroup favoritism [29,30]; if individuals perceive gender-based initiatives as generally beneficial to women (as a group) but not men, and they are motivated to act in ways that support their own gender-based ingroup (e.g., if they highly identify with their gender), women may be generally supportive of these initiatives while men may not be, especially if men's overestimation of women in the field helps justify a belief that making deliberate efforts to support members of an outgroup are no longer necessary (i.e., supporting initiatives that perceptibly benefit women)."

In terms of the literature review, I was unclear about how this study fits into the existing literature regarding estimates of women's representation in medicine. Has there been past research on the overestimation of women in medicine, or is this study the first? What about past research on estimations of women in other professions? I would recommend adding a literature review on the effects of biased estimates of group representation.

Response #8

We really appreciate the encouragement to dig a bit further into the existing literature on this topic. Unfortunately, to our knowledge, previous work on this topic is quite limited.

In our original version of this manuscript, we briefly described relevant findings from Swim et al., 1995 (*JPS*). Supplementing this, we now cite a couple of additional papers, which similarly indicate that overestimating

women's representation (e.g., in politics) predicts lower support for initiatives that aim to support women and/or greater gender equality in those fields (Coffé & Reiser, 2021; Sanbonmatsu, 2003).

In efforts to keep our manuscript as close as possible to advised word limits, and given the consistent patterns of findings across these few studies, we have opted not to create a separate subsection discussing this small bit of literature. However, in addition to citing these additional papers, we have revised our manuscript to make it clearer that previous literature on this topic is indeed rather limited. Specifically, we now state (new text in bold):

“Indeed, **previous research on this topic, though limited in scope**, demonstrates that when individuals overestimate women's representation in a field (e.g., in **Science, Technology, Engineering, Mathematics and Medicine [STEMM], politics**), they show less support for initiatives that aim to help women in those fields [9–11]. Thus, medical professionals who overestimate...”

Overall, I found the Methods section very clear. I do feel the paper would benefit from definitions of each type of doctor in the study – for example, I am unclear about the distinction between general practice and medicine, or junior doctors and consultant doctors.

Response #9

In efforts to keep our manuscript as close as possible to advised word limits, we have made a concise revision that helps direct interested readers to this information (on websites maintained by the British Medical Association and the UK's National Health Service) by stating (in Participants and Procedure; all text is new):

“For more detailed descriptions of these areas and roles within medicine, see [18,19].”

In terms of Results, I think that Table 2 should be broken out by the gender of respondent. I was very curious about whether men or women were more likely to overestimate women's representation in each area. I would also recommend breaking out the data by age of respondent and by respondent's role – in particular whether they are a manager or a supervisor. There are potentially greater implications for the biased estimations of people in positions of power (who make hiring decisions, etc.).

Response #10

We appreciate these suggestions. While we do not have data on respondents' status as managers or supervisors, we do highlight the value of this idea for future research – see Response #12. We also feel that a breakdown of mean (over)estimations by age would be rather cumbersome (~50 discrete ages in the data * 7 discrete areas/roles = 350 *t*-tests), though because these data are publicly accessible one could certainly explore these questions if they wanted to.

At the same time, in line with your recommendation to break out the results by gender of respondent, which seems particularly relevant to the current study, we now provide this more detailed information in a new Table 3 (similar to Table 2 in layout; note that we found it rather cluttered and 'busy' when trying to put these additional gender-specific results directly into Table 2).

As can be seen in Table 3, patterns of (over)estimation held consistently across male and female respondents. When directly comparing the magnitude of male and female respondents' mean deviations from the true representation of women in a given area/role (i.e., whether male and female respondents' mean [over/under]estimations for a given area/role significantly differed from one another), results showed that, by and large, the magnitude of these deviations did not significantly differ (exceptions, as noted in Table 3 with superscript letters: female respondents' mean overestimation of women consultants/GPs in General Practice was larger than that of male respondents' [$p =$

.01, $d = .26$]; male respondents' mean *underestimations* of women trainees in General Practice was greater than that of female respondents' [$p = .03$, $d = .22$]).

I am also interested in why the researchers think gender stereotypical beliefs did not correlate with overestimations – that seems counterintuitive so it would be good to have more discussion of this finding.

Response #11

Thanks for raising this point, which we agree is an interesting one. We now briefly discuss this in the Discussion.

Specifically, we state (all text is new):

“It is also notable that medical professionals' endorsement of the gender-stereotypical belief that men are better suited for the profession was unrelated to their tendency to overestimate the proportion of women in the field (see Table 1). This held true for both male and female respondents. It suggests that overestimations of women's representation do not simply reflect a negative, pre-existing attitude (about women's suitability for the profession). Thus, while future research should further probe this relationship, their independence here indicates that medical professionals' estimates of women's representation are, in their own right, an important basis for understanding who is likely to support gender-equality initiatives, or resist them – particularly among men in the profession. While endorsement of this gender-stereotypical belief is important to consider, medical professionals' (over)estimations of women is key too.”

In terms of future directions, I would be interested in the researchers' views on what the data would look like if the respondents were from the general public, instead of medical professionals. What are the implications of the public overestimating the percentage of women in medicine vs. medical professionals' overestimation?

Response #12

Another intriguing question, thanks! While we refrain from giving too much space to future directions and open-ended questions, we do now briefly discuss the value of examining these processes in other potentially important populations, including among the general public, and among leaders within the medical profession specifically (relating to an earlier comment; see Response #10). Specifically, we now state (all text is new):

“Future research might also examine whether the general public similarly tends to overestimate women's representation in the medical profession. Individuals outside the profession would presumably be just as prone, if not more so, to these erroneous estimates. If so, given the current evidence that this has adverse implications for one's willingness to support gender-equality initiatives, this would underscore the gravity of the issue – highlighting that resistance to establishing gender equality in the medical field may be coming from both those within and outside of the profession. In a similar vein, it will be valuable to examine whether these processes are evident specifically among leaders within the medical profession.”

There are of course a host of other potential implications we could discuss. For instance, among the general public, might overestimating women's representation in the medical field prompt a general devaluation of the field? Or might it expedite the normalization of, and thus decrease resistance to the idea of having/being assigned, a female surgeon for example? However, with this new text we opted to stay closer to the current study's findings and the implications that directly stem from it (regarding support for gender-equality initiatives).

Last of all, some minor points:

- STEM should be written out the first time it is mentioned

Response #13

We have now written this out when it is first used.

- In the first line of the current research section – “The current research examines medical professionals’ tendency to overestimate . . .” – I would add the word “whether” – the introduction shouldn’t be written as if overestimation is a given

Response #14

Good point, thanks for noting it. That sentence has been revised to now state, “The current research examines **whether** medical professionals’ tend to overestimate women’s representation in medicine...”

References:

Crawley, D. (2014). Gender and perceptions of occupational prestige: Changes over 20 years. Sage Open, 4(1), 2158244013518923.

Reviewer: 3

Dr. Amy H Farkas, Medical College of Wisconsin

Comments to the Author:

This is a very interesting paper that highlights the role that perception versus reality plays in gender equity within medicine. This is a very well done study. The paper is well written with a clear description of their methodology and results. The discussion highlights the importance of this work. My only suggestion to the authors would be in their discussion they state that multiple strategies might be needed in order to overcome the issue of overestimation and I wonder if they could suggest some possible strategies to foster support for gender equity programs. Beyond that I do not have any additional suggestions.

Response #15

Thanks so much for your supportive and helpful comments. As requested, we have extended our General Discussion to say more about what strategies may be necessary to overcome this issue of overestimation, and to foster support

for greater gender equality in the profession more broadly. While making efforts to keep our revised text concise, we have also added references to articles that provide further guidance and information about potential strategies for addressing persisting issues of gender inequality. Specifically, we now state (new text in bold):

“Overall, this suggests multiple strategies may be required to address the consequences of this overestimation effect, depending on whether or for whom it is underpinned by a sense of threat versus naïveté about ongoing issues of underrepresentation (if not also ongoing issues of gender bias). **For instance, targeted information campaigns that increase knowledge and awareness about women’s true representation in different areas of medicine – along with information about persisting forms of gender bias (separate from matters of representation) – may be useful in fostering greater support for gender-based initiatives among medical professionals whose reservations about these initiatives are rooted in genuine naïveté about persisting issues with underrepresentation and bias. Yet among those whose resistance is rooted in a sense of threat by growing proportions of women in the profession, other strategies may be necessary (e.g., work-related self-affirmation techniques that alleviate this sense of threat) [25,26]. There are a number of other potential strategies to consider as well, including those that aim to directly promote greater gender equality (for reviews, see [14,27]).**”

Reviewer: 4

Prof. Elizabeth Holliday, The University of
Newcastle, Australia
Comments to the
Author:

This is an interested and well written paper with fascinating conclusions. If these conclusions are valid, this paper would represent an important addition to the literature.

While I greatly enjoyed reading the introduction, my review of the methods and results was hampered by the very limited description of statistical methods used. This made it difficult to understand how the analysis was performed, and how the results should be interpreted. However, I believe these shortcomings can be readily rectified and have outlined suggestions for doing so below. I am willing to review a revised version of this manuscript that addresses my suggestions.

Methods

I found the reporting of statistical methods to be incomplete; I could find no “statistical methods” section in the methods. Please include such a section that describes all statistical tests used, and the software package. Then update the STROBE checklist to show where this content is provided.

For example, please the type of correlation statistic and how the correlation was estimated to assist with interpretation of results in Table 1 (or Table 2, as per my suggestion for a new Table 1 below).

Response #16

Thanks so much for your supportive and helpful comments. We have now moved our descriptions of statistical approaches to a single subsection at the end of the Methods (see

pg. 9 of the manuscript), and have expanded these descriptions to provide more detail, as requested (e.g., specifying that Table 1 reflects bivariate [zero-order, Pearson] correlations). We have also updated the STROBE checklist to reflect these and all other revisions.

The meaning of the correlation values in Table 1 was unclear to me – which variables do these represent the correlation between?

Response #17

As noted above, these reflect bivariate (zero-order) correlations between pairs of variables, estimated separately for female respondents (above the diagonal) and male respondents (below the diagonal). The numbering across the top row of the table (1-9) correspond to the variables, as numbered, in the left column. To ensure that this is clear to readers, we have added and/or reiterated these details. Specifically, we have now expanded the Table 1 title to offer more detail, expanded the Table 1 notes to clarify what the numbering across the top row of the table represents, and have added a new paragraph preceding Table 1 that describes the basic contents of the table.

On this same point, later in the results section there are interaction estimates presented, with confidence intervals and model fit statistics. Please describe the regression model used to estimate these terms, specifying the response variable and all explanatory variables. Also describe how any assumptions of the model were assessed. It is difficult to properly assess the results without this information. I later noticed a brief description of a moderated regression analysis mentioned partway through the results section, but this content should be described in the methods, after first specifying the type of regression model used. Please describe the type of regression model using conventional terminology, that makes the assumed distribution for the response variable or errors clear, e.g., linear regression or a generalised linear model with the assumed response distribution and link function used. For the interaction terms, I suggest you describe assessing “interaction” or “effect modification” rather than “moderator analysis” for consistency with the rest of the paper and with terminology widely used in epidemiology.

Response #18

We have now provided much of this detail in a new subsection at the end of Methods, where we provide an overview of statistical methods. Specifically, we now state (all text is new):

“...tests of interactions using linear (ordinary least squares) regression via the PROCESS macro in SPSS, with 5,000 resamples for generating percentile bootstrap confidence intervals (for more details about PROCESS, see [23]). Primary regression analyses tested whether respondents’ support for gender-based initiatives varied as function of their tendency to overestimate the proportion of women in medicine and their own gender (overestimation*respondent-gender interaction) using PROCESS Model 1 (outcome: support for gender-based initiatives; predictor: overestimation of women’s representation [mean-centered]; moderator: gender [0 *female*, 1 *male*; mean-centered]; covariate: age; analyses without covariate evinced the same statistically significant results). Follow-up regression analyses mirrored primary regression analyses while further testing whether the hypothesized overestimation*respondent-gender effect was robust and/or qualified by respondents’ endorsement of the gender-stereotypical belief that men are superior for the medical profession (overestimation*respondent gender*gender-stereotypical belief) using PROCESS

Model 3 (regression model identical to the primary regression model, but with the inclusion of a second moderator, endorsement of gender-stereotypical belief, and its corresponding interaction terms).”

As can be seen, following your suggestions, we now provide more detailed descriptions of statistical methods, models for testing moderation, and have revised the ‘moderation’ term to align with descriptions of corresponding results (now referred to as ‘tests of interactions’). Regarding underlying model assumptions, following more recent standards in psychology (if not other fields), we employ methods that avoid sole reliance on traditional model assumptions. Most notably, this includes our use of resampling (with replacement) to generate percentile bootstrap confidence intervals, which do not require distributional assumptions (e.g., normally distributed errors). As an aside, in our experience it can be useful to note that, “Contrary to the beliefs of some, the assumption of normality does not pertain to the distribution of Y itself [the outcome variable] or to the predictors of Y in the regression model. Regression analysis makes no assumptions about the shape of these distributions” (Hayes, 2018).

As can be seen in the newly added text stated above, we make clear to readers that we use percentile bootstrap confidence intervals. With advised word limits in mind, we do not get into too much detail about this in the manuscript but would like to also note here that to further probe the potential of heteroscedasticity influencing our findings, we reran our primary regression analyses (tests of interactions, with 5,000 resamples) using an alternative, heteroscedasticity-consistent covariance matrix estimator (HC3; see Hayes & Cai, 2007). Results evinced all the same statistically significant results as in our original analyses. We also reran our *t*-tests using 5,000 resamples for producing percentile bootstrap confidence intervals. Again, we found all the same (statistically significant) results as in our original analyses. Finally, while there are differing perspectives on how to handle potential outliers – and with our sample size, this seems like a relatively minor concern – we did inspect our data for outliers on any key variables, via boxplots that specify the median, the IQR (computed from Tukey’s hinges), and any cases with values outside that IQR by a factor of three (this is a standard metric used in SPSS for identifying extreme outliers). When removing these potential outliers ($n=8$), all of our findings remain the same. In fact, if anything, our effects get stronger (smaller *p*-values, larger *d*’s).

Results

Tables

It is standard to include a table showing participant characteristics for key variables that may relate to the exposure or response, e.g., as frequency with percent for categorical variables, and mean with SD or median with IQR for continuous variables. I suggest including this as a new Table 1 and renumbering the existing tables to follow.

Response #20

Thanks for this suggestion, and we appreciate that different disciplines have different standards for how information about participants and key variables are reported (e.g., in text vs. tables). In this manuscript, we sought to ensure that this basic information is provided yet while avoiding too much redundancy across text and tables. As such, we have opted to provide basic demographic information in text, and basic information about key variables in relevant tables (e.g., Table 1 provides an overview of bivariate correlations, Table 2 provides means and SDs, and Table 3 now provides a breakdown of these means and SDs by

respondent gender). And of course, because we have made these data publicly accessible, any additional information desired can be gathered by interested readers.

When describing results from the regression models, it would be helpful to offer the reader explicit interpretations of the beta coefficients and particularly the interaction terms at key instances (e.g., one explanation for each model). In addition to clearly describing the models (including outcomes and explanatory variables) in the methods, such interpretations will help the reader to understand the magnitude and relevance of the estimated effects. This is particularly important for interaction terms, which most readers do not intuitively understand. Spell it out – what does a positive or negative, significant interaction term mean in the context of each model? What does it imply about the modification of a particular effect by gender, for example?

Response #21

Thanks for this suggestion, and we agree that regression coefficients are not always easily interpreted. Ultimately, we believe the coefficients affiliated with the simple slopes are the most important and relevant for readers to understand. Therefore, we have now added a more thorough and straightforward interpretation of these coefficients (and similarly so for our follow-up regression analyses that probe a three-way interaction).

In particular, when describing the primary interaction (see Figure 1), we now state (new text in bold):

“...simple slopes further showed that female respondents’ (over)estimates were unrelated to their level of support ($B=.00$, 95% CI $=-.02$ to $.02$, $p=.92$), yet male respondents’ tendency to overestimate the proportion of women in medicine predicted lower support for gender-based initiatives ($B=-.04$, 95% CI $=-.06$ to $-.02$, $p < .001$). **In other words, among female respondents, regardless of their estimations of women in medicine, there was no systematic difference in their level of support for gender-based initiatives. Yet among male respondents, there were systematic differences; in essence, for every 1% increase in their (over)estimations of the proportion of women in medicine, men’s support for gender-based initiatives dropped by .04 points on average (thus, being 12% higher in one’s overestimations equated to approximately a half-point decrease in level of support; see Figure 1 for a visual illustration).**”

We believe that this newly added text, along with the visualization of results (Figure 1), will help readers better understand our findings in a meaningful way. To supplement this, we have also added a more general description of the interaction. Specifically, just before describing the simple slopes, we now state (all text is new):

“Generally speaking, this means that as medical professionals got more severe in their overestimations of women’s true representation, the disparity between female and male medical professionals’ support for gender-based initiatives grew larger – as illustrated in Figure 1.”

Notably, we have not offered more detailed descriptions of the interaction or other coefficients in the manuscript (though see below) in part because a direct interpretation of them is, in itself, probably not that useful to readers. For instance, the interaction term does not have a particularly intuitive meaning (in essence, it describes a difference between differences). Similarly, the two coefficients affiliated with the two key predictors in the regression model (respondent gender, [over]estimated proportion of women in medicine) are not very meaningful in their own right

– in part because these effects are qualified by an interaction, and in part because the coefficients themselves are shaped by the (largely arbitrary) coding of the gender variable in the data (0 *female*, 1 *male*), along with the use of mean centering in analyses (though mean-centering is generally valuable for ensuring coefficients reflect meaningful *values* within the data). Nevertheless, as a little FYI, here is a description of these coefficients (for our primary analyses, as illustrated in Figure 1):

Overestimation, $B = -.02$: When the mean-centered equivalent of respondent gender is 0, for every 1% increase in individuals' (over)estimation of the proportion of women in medicine there is an average drop of .02 points in their level of support for gender-based initiatives. Note that because of how gender is coded (0 *female*, 1 *male*), and the slight difference in the numbers of female and male respondents in the sample (sample is ~47% female), when the mean-centered equivalent of respondent gender is 0, this is roughly, but not exactly, analogous to saying that this coefficient is an applicable depiction 'across the two genders' – though again, this is not precisely correct, given that there is not exactly the same number of female and male respondents.

Respondent gender, $B = -.40$: At the mean of the sample's (over)estimation of the proportion of women in medicine (i.e., at the 'average' level of overestimation within the sample, or when the mean-centered equivalent of the sample's overestimation is zero), compared to female respondents, male respondents are on average .40 points lower in their level of support for gender-based initiatives. As can be seen, this coefficient only describes the size of the gender difference (in level of support for gender-equality initiatives) at one particular level of (over)estimation (i.e., at the mean level of overestimation within the sample). It may also be worth noting that if we had not used mean-centering in our analyses, this coefficient would look different and arguably be less meaningful as it would describe the size of the gender difference at the point where individuals neither over- nor underestimate women's representation (rather than describing the size of the difference at the mean level of overestimation). Moreover, as the significant interaction term indicates, this value is not particularly important because the size of this gender difference systematically changes as a function of just how much individuals overestimate women's representation.

Overestimation*respondent-gender interaction, $B = -.04$: For every 1% increase in individuals' (over)estimations of the proportion of women in medicine, the *magnitude of the difference* between female and male respondents' level of support for gender-based initiatives grows by .04 (as you are probably aware, the sign of the interaction term [negative vs. positive] reflects the particular coding of the gender variable; Figure 1 provides a nice supplement to illustrate the nature of this interaction).

Our sense is that the more general statement of the interaction term, which we have now added (see new text quoted earlier in Response #21), along with the illustration of results in Figure 1, and the expanded description of simple slopes (see new text quoted earlier in Response #21), will together provide readers with a meaningful understanding of our findings.

VERSION 2 – REVIEW

REVIEWER	Ni, H. Wenwen Sonoma State University, Psychology
REVIEW RETURNED	25-Jan-2022

GENERAL COMMENTS	Great work!
-------------

REVIEWER	Farkas, Amy H Medical College of Wisconsin, Division of General Internal Medicine
REVIEW RETURNED	10-Jan-2022

GENERAL COMMENTS	This is an important and well done manuscript. I think the authors have addressed the few comments from the reviewers well and I have no further suggestions.
---

REVIEWER	Holliday, Elizabeth The University of Newcastle, Australia, Rm4107, Lvl4W
REVIEW RETURNED	14-Jan-2022

GENERAL COMMENTS	The authors have done a commendable job of addressing my original concerns. I find the revised manuscript has been greatly improved by their addition of substantial text outlining the statistical methods and interpreting the results.
---